# Morphometric Characterization of Local Goat Breeds in Two Agroecological Zones of Burkina Faso, West Africa

**DOI:** 10.3390/ani13121931

**Published:** 2023-06-09

**Authors:** Badjibassa Akounda, Dominique Ouédraogo, Albert Soudré, Pamela A. Burger, Benjamin D. Rosen, Curtis P. Van Tassell, Johann Sölkner

**Affiliations:** 1Unité de Formation et de Recherches Sciences et Technologie, Université Norbert ZONGO, Koudougou BP 376, Burkina Faso; ambrorione@gmail.com (B.A.);; 2Centre Universitaire de Ziniaré, Université Joseph KI-ZERBO, Ouagadougou 03 BP 7021, Burkina Faso; 3Research Institute of Wildlife Ecology, University of Veterinary Medicine Vienna, Savoyenstrasse 1, 1160 Vienna, Austria; 4Animal Genomics and Improvement Laboratory, United States Department of Agricultural, 10300 Baltimore Ave, Beltsville, MD 20705, USAcurt.vantassell@usda.gov (C.P.V.T.); 5Division of Livestock Sciences, Department of Sustainable Agricultural Systems, University of Natural Resources and Life Sciences Vienna, Gregor-Mendel Strasse, 1180 Vienna, Austria

**Keywords:** characterization, body measurements, goat, breeds, Burkina Faso, West Africa

## Abstract

**Simple Summary:**

Phenotypic variability within and between populations can be assessed using morphometric traits. Therefore, this study investigated morphological differences between different goat populations in two distinct agroecological zones of Burkina Faso. Based on 18 linear body measurements and body weight of the 511 adult female animals sampled, no strict distinction was found between the two populations, meaning that these populations are genetically close. Based on body size, four groups were identified representing the two main breeds expected in these areas (Djallonké and Mossi), their crosses, and the crossbred between Mossi type and the third main breed (Sahelian type) of the country. These different goat morphotypes are the results of farmers breeding practices, specifically uncontrolled mating but also intentional crossbreeding in the search for traits of interest.

**Abstract:**

In Burkina Faso, goats are the second most numerous ruminant livestock population, with almost exclusively indigenous breeds being reared in extensive production systems in various agroecological zones. This study was carried out to understand the morphological variation of local goat breeds in the Sudano-Sahelian and Sudanian agroecological zones. A total of 511 adult female animals belonging to two presumed populations (Mossi breed in Sudano-Sahelian zone and Djallonké breed in Sudanian zone) were sampled and body weight as well as a range of linear body measurements, following FAO guidelines, were recorded. The least squares means of body measurements of indicated that Sudano-Sahelian goats have significantly (*p* < 0.001) larger body measurements than Sudanian goats. Furthermore, relative high variability of the two populations in morphometric traits was observed. Principal Component Analysis (PCA) suggested structure between Mossi breed on one side and Djallonké on the other side, but no strict separation was observed, suggesting that gene flow is occurring among the different populations. A dispersion map with four clusters was built based on the first two factors. The least square means of body measurements ranked the four groups from small to large body size, namely Djallonké, Mossi × Djallonké, Mossi, and Sahelian × Mossi. Gene flow from Sahelian goat into other populations of the country, based on migration of the Fulani ethnic group from the Sahel into areas with Mossi and Djallonké breeds, could explain this configuration and confirms the continuous erosion of genetic identity of these two local breeds. The sustainable use of these adapted local goat genetic resources calls for the promotion of sustainable genetic improvement using participatory breeding approaches.

## 1. Introduction

In Burkina Faso, goats are the second largest ruminant population. They are exclusively indigenous breeds reared in extensive production systems in various agroecological zones. Indeed, with nearly 15.5 million animals, goats represent about 43% of the domestic ruminants herd, putting Burkina Faso among the countries hosting an important population of this species with a particular socioeconomic importance [1]. Goats provide diverse goods and services to the majority of the population living in rural areas, including food, income, and employment, as well as being living saving accounts, insurance, and important for sociocultural needs [2,3,4,5,6].

Three main goat breeds are traditionally recognized in Burkina Faso based on ethnic and/or geographic location nomenclatures. The Sahelian breed, characterized by a large and elongated body with long and thin limbs, is distributed throughout the Northern region. The Djallonké breed, located in Southern Burkina Faso (Sudanian area), is a short-eared, small-horned goat belonging to the West African dwarf goat population. In the Sudan-Sahel area of the country, there is an intermediate breed between the two previous, named the Mossi breed [7,8,9]. Burkina Faso goat breeds were previously characterized both phenotypically based on body measurements and genotypically using microsatellites [6,7,9]. However, the state of knowledge regarding the breeding practices is still limited, causing continuous introgression and removing distinct boundaries between these breeds. Indeed, the genotypic variation in a population arises from evolutionary forces such as mutation, drift, selection, and migration that result in changes of allelic frequency in space and time. However, the magnitude in phenotypic variability and distance could differ under different environmental conditions and farming practices [10].

The characterization of African small ruminant populations will play a major role in the maintenance of these autochthonous genetic resources as the basis for future improvement at both the production and the genetic levels [10]. This characterization can be achieved through analyses of morphological variation of animals within and between populations. Characterization of livestock breeds based on their morphological variation is a first step toward the inventory of Animal Genetic Resources (AnGR) and is a prerequisite for visualizing the diversity of animal genetic resources and determining their level of genetic erosion [11]. Animal morphological measurements have diverse implications in breeding perspectives. Body measurements such as chest girth, height at withers, and body length can be used for the rapid choice of large size individuals in the field to enable the establishment of elite flocks [12]. Selection and breeding based on morphometrical measurement can result in the improvement of productive traits such as live weight of indigenous goats for meat production [12,13].

Therefore, the aim of this study is to contribute to the genetic characterization of native goats by investigating their morphological differences in different agroecological zones of Burkina Faso. The results of this study may be useful and important for future breeding programs and management plans of these local breeds.

## 2. Materials and Methods

### 2.1. Agroecological Characteristics of Survey Areas

The study was carried out in two provinces, Namentenga and Poni (Figure 1), corresponding to two agroecological zones of Burkina Faso: the Sudano-Sahelian zone (ZSS) and the Sudanian zone (ZS) according to the phytogeographic division of the country by Guinko [14]. Namentenga province is located between 11°3′ N and 13°5′ N latitudes while Poni province is situated from 9°3′ N to 11°3′ N. The Sudano-Sahelian zone is characterized by a short rainy season from June to October (4–5 months) and a long dry season (7–8 months). In the Sudanian zone, the rainy season lasts from May to October (5–6 months) [15]. With the current climate fluctuations, these characteristics have undergone deep changes marked by the north–south sift of climatic isohyets, the decline of Sudanian band, the decline and irregularity of rains, and the reduction of vegetation cover. At demographic level, these changes were accompanied by the movements of Sahelian and Sudano-Sahelian peoples with their animals into to Sudanian zones searching for crop land and grazing lands. Similar to cattle, introgression of Sahelian goat breeds into to Sudanian ones occurs due to breeding practices and the migration and settlement of Fulani people with their herds from Northern to Southern regions [6,9]. Thus, Sahelian and Mossi types are most likely to be found in the two areas while Djallonké type animals are only expected in the Sudanian zone.

### 2.2. Animal Sampling and Body Measurements

A total of 511 female goats with a minimum age of 3 years in five sites were sampled in the two provinces. The number of animals sampled per site is reported in Table 1. A maximum of 4 animals in each herd were used for body measurement in order to avoid sampling of related animals. Nineteen (19) quantitative traits were assessed according to the FAO/ISAG guidelines for goat characterization [16]. Age of animals was estimated through dentition examination and animals with 4 and more permanent incisors (PPI) were considered to be above 39 months i.e., at least 3 years [10]. The following morphological traits were measured and recorded: height at withers (HW, the distance from the surface of a platform on which an animal stands, to the withers of the animal); height at rump (HR, the vertical distance between the ground and the point determined by the intersection between the line passing through the points of the hips and the rump); height at back (HB, the distance from the surface of a platform on which animal stands, to the back of the animal); chest girth (CG, the circumference of girth); chest depth (CD, the distance from the brisket between the front legs to withers); body weight (BW, the live body weight of the animal); posterior and anterior cannon length (PCL and ACL, the length of posterior and anterior cannon bone); body length (BL, the distance between the point of the shoulder and the pin bone); Ischium width (IW, the distance between tuber ischii); shoulder width (SW, the distance from the left to the right shoulder blade); Tail length (TL, the distance from the tail droop to tip of the tail excluding switch); neck length (NL, the distance from the head to the thorax); rump length (RL, the distance from the hip (tuber coxa) to the pin (tuber ischii) by dividers); ear length (EL, the distance from the base to the tip of the ear along the dorsal face); ear width (EW, the maximum distance at the middle of ear); horn length (HorL, the distance between the base to the tip of horn along the greater curvature); chest width (CW, the distance between the axis of the forelimbs at the base of the sternum) and head length (HL, the distance from the bun to the middle of muffle). The live body weights of the goats were taken using a suspended weighing scale (200 kg capacity with 100 g precision) and other body linear measurements were measured with tape calibrated to centimeters (cm) and self-devised scaling sticks made from iron. For this purpose, animals were restrained and held in natural position.

During the process of data collection, animals were assigned based on their phenotypic appearance to genotypes corresponding to the local breeds of the country named Djallonké, Mossi, Sahelian, and their crosses by the research team. This assignment has been done based on the experience of the team from previous studies.

### 2.3. Data Processing and Statistical Analysis

The data was registered directly on the Online Data Kit (ODK)/KoboToolbox platform, then exported to Excel and transferred to R software [17] for statistical analysis. Descriptive statistics were performed for the quantitative linear body measurements and body weight. The least squares means (LSmeans) of the morphometric traits, their standard errors (SE), and related coefficients of variation (CV) were estimated for each agroecological zones. The discrimination of goats from their morphometric characteristics was made using different discriminant analysis. First, a hierarchical classification on principal components (HCPC) was performed with nine (09) variables, including 8 body measurements and the agroecological zones which were strongly correlated with the first two axes of the PCA. The 8 body measurements used were 7 linear body measurements, which are body length (BL), height at withers (HW), height at back (HB), height at rump (HR), chest depth (CD), chest girth (CG), posterior cannon length (PCL), and body weight (BW). The others body measurements were removed to improve the total inertia because they weakly contributed to the first two dimensions of the PCA. As well, the number of morphotypes existing within the local goat population of Burkina were also identified using ascending hierarchical classification (AHC). The quality clusters obtained were assessed using average silhouette width and permutation test. Clusters were considered significative if the average silhouette observed for real clusters was significantly higher than those of clusters generated by permutation. The morphotypes observed were then characterized by one-factor variance analysis (ANOVA) (morphotype or identified group): Comparisons between agroecological zones and between morphotypes were done using Student–Newman–Keuls (SNK) mean structuring tests. The level of significance of 5% was used to compare the parameters of the different groups. A linear discriminant analysis was finally performed to assess the discrimination of genetic groups from the morphometric variables. This classification was then compared to the subjective expert assignment. The conditions of application of all the statistical tests used had been verified beforehand.

## 3. Results

Morphometric traits variation among the two agroecological zones are presented in Table 2. The results indicated significant (*p* < 0.001) variations among the two zones for all linear body measurements. The highest mean values of body weight and most of the linear body measurements were observed in the Sudano-Sahelian area. The coefficient of variability (CV) indicated that variability ranged from 6.31% (height at rump) to 27.48% (chest width) in the whole population. Globally, CV values were relatively low for most of the traits except for body weight (BW), ischium width (IW), tail length (TL), and Muzzle circumference (MC), which were 18.63%, 14.87%, 12.02%, and 27.48%, respectively.

Relationships between traits were analyzed by constructing bidimensional dispersion plots. The first two components of the PCA plot accounted for 85.4% of the total variability of the data (Figure 2). Eight traits contributed mostly to the morphological variability in the studied populations (Height at rump, height at withers, height at back, chest girth, body length, body weight, post canon length, and chest width). All these variables were positively and strongly correlated (>0.6) to the first principal component (PC1), which accounted for 73.3% of the total variance, while PC2 accounted for 12.1% and had positive correlations with body weight, body length, chest girth, and chest depth and negative correlation with the remaining traits. The graph of distribution of the variability of individuals showed four main clusters: the Djallonké breed in the Sudanian area, the Mossi breed in the Sudano-Sahelian zone, and the crossed (Mossi × Djallonké and Sahelian × Mossi) in both agroecological zones (Figure 3). However, overlapping clusters indicated that there is no strict separation between the populations of the two agroecological areas. Referring to individual characteristics, the first dimension of the PCA graph of individuals showed that the four populations have different morphological shapes. On one hand, the groups of Sudano-Sahelian zone are characterized by high morphological values (body length, height at the croup, height at the withers, length of the ears, thoracic circumference, and weight), and on the on other hand, the populations of Sudanian area are characterized by low body measurements. Between the two groups in the overlapping area of the clusters appear groups with intermediate performances.

The results of the ascending hierarchical classification optimized by the determination of the possible number of groups (K) suggested that the optimal number of groups was four (Figure 4). The average silhouette withs were 0.27, 0.25, 0.28, and 0.22 for clusters 1, 2, 3, and 4, respectively, suggesting a good quality of clustering (Table 3). Furthermore, permutation test showed that average silhouette width for real clusters was significantly (*p* < 0.001) higher than those generated by permutation, confirming the validity of our clusters. According to the least squares means of the seven linear body measurements and body weight used for the clustering, the four groups can be arranged from small to large body size (Table 4) and assigned to genotypes, namely, Djallonké, Sahelian × Djallonké, Mossi, and Sahelian × Mossi regarding the history of goat breeds in the sampling areas. Indeed, the least squares means and standard errors of linear body measurements and body weight of the Djallonké (Group 1) were 53.34 ± 3.02 cm, 45.12 ± 2.52 cm, 44.09 ± 2.31 cm, 45.81 ± 2.33 cm, 20.77 ± 1.04 cm, 57.99 ± 2.68 cm, 13.77 ± 0.89 cm, and 16.87 ± 1.96 kg, respectively, for body length, height at withers, height at back, height at rump, chest depth, chest girth, posterior cannon length, and body weight. The corresponding values for the Sahelian × Djallonké (Group 2) were 57.18 ± 2.26 cm, 47.47 ± 2.19 cm, 46.23 ± 1.78 cm, 48.10 ± 2.14 cm, 22.62 ± 1.32 cm, 62.58 ± 2.46 cm, 14.12 ± 0.97 cm, and 21.25 ± 2.22 kg. The Group 3 (Mossi) showed 58.81 ± 2.70 cm, 53.88 ± 2.41 cm, 52.58 ± 2.47 cm, 54.22 ± 2.74 cm, 23.13 ± 1.24 cm, 64.08 ± 2.27 cm, 16.29 ± 0.93 cm, and 22.00 ± 2.51 kg for body length, height at withers, height at back, height at rump, chest depth, chest girth, posterior cannon length, and body weight, respectively. The corresponding measurements were 63.16 ± 2.38 cm, 57.09 ± 3.07 cm, 54.75 ± 3.19 cm, 56.59 ± 3.18 cm, 25.03 ± 1.20 cm, 68.43 ± 2.75 cm, 16.91 ± 1.08 cm, and 27.51 ± 3.60 kg for the Group 4 (Sahelian × Mossi). All the groups were significantly (*p* < 0.05) different for all these linear body measurements and body weight.

A comparison of assignments by hierarchical classification and subjective evaluation of animals provided extremely high concordance. Indeed, the assignment was 100%, 96.27%, and 88.93% true for Djallonké × Mossi, Mossi, and Djallonké, respectively (Table 5). However, for the group predicted as Sahelian × Mossi by the model, the assignment based on visual observation was true only in 63.64% of the cases, the other 36.36% were Mossi.

## 4. Discussion

Morphological measurements are a useful tool to describe variability within and between populations. Therefore, they have been used for the characterization of many livestock populations worldwide [18,19,20,21,22,23,24,25,26,27,28,29]. Male animals were excluded from the analysis of the current study due to their low number at adult age. Previous investigation on West African goats morphological characteristics were also done only with female animals [8,30,31]. The low proportion of adult males in flocks could be explained by several reasons. Male animals are preferentially used in sociocultural activities and firstly sold in case of cash incomes needs. Furthermore, unlike sheep, there is no fattening practice with goats in Burkina Faso. The lack of adult males in goat herds has already been reported in Burkina Faso by previous studies [7,29]. Lack or low number of breeding males in flocks can lead to low sex ratio and this could have implications for genetic diversity and for the implementation of breeding programs.

The findings of this study suggest notable variation in the physical characteristics of goats in Burkina Faso. Overall, the goats in the two agroecological zones showed high variability in morphometric traits. Globally, Sudano-Sahelian population showed large body proportions compared to goats in Sudanian zone, suggesting distinct ancestry of goats in the two areas. This trend corroborates the findings of previous similar studies of goats in Burkina Faso and in Benin [8,30,31]. Compared to the findings of previous studies, except of body length and horns length which were higher, measurements such as height at withers and ear length of Sudanian goats in this study were close to the findings of [29] in the same area. However, except for height at rump and horns length, other body measurements of our study, such as height at withers, height at rump, chest depth, body length, ear length, neck length, tail length, chest girth, and body weight, were lower than those of Guineo-Sudanian goats of Benin [31]. However, these two population were suggested to be nearer [30]. For Sudano-Sahelian goats, our results for body length, height at withers, horns length, and ear length were higher than those of a previous study in the same area [29]. While Sudano-Sahelian goats of Burkina Faso are suggested to be close to Sudanian goats of Benin [30], except for height at withers, height at rump, and neck length, their other body measurements were lower than Benin Sudanian goats. All these differences could be explained by the difference in farmers’ breeding practices, mainly the crossbreeding between different genotypes and also by the natural adaptation process. Indeed, body size and shape are the most dominant morphological characteristics influencing the adaptation of animal to harsh environments [10,32]. The coefficient of variation (CV) values in this study were relatively low for most of the traits except for chest width, ischium width, tail length, and body weight. These CV values remained lower than those found by [7] but globally in the same range of those of [31].

Principal Component Analysis (PCA) suggested structure between Mossi and Djallonké goats and the lack of strict separation, suggesting that genes flow is potentially occurring between the two agroecological zones and specifically from the Sudano-Sahelian into the Sudanian populations. High gene flow in livestock populations in West Africa has been suggested to be a consequence of several factors including mobility and breeding practices. The high phenotypic variation observed in this study would be linked to the action of nomadic Fulani people who move with their Sahelian breeds from the north into the south in search of pasture, causing crossbreeding with native Sudanian types due to uncontrolled mating [30]. In addition, the need of farmers for animals with specific features and specifically large body size leads to crossbreeding between the bigger Sahelian breeds and the smaller Sudanian breeds. Crossbreeding has already been pointed out as one of the reasons of introgression of Sahelian genes into Sudanian animals [9,22,33]. Previously, ref. [30] reported that anthropological factors, including farmers’ breed preference and consumers’ demand, as well as the frequent mobility of some herdsmen with their animal, favor crossbreeding. However, Djallonké and Mossi breeds, due to their relative low productivity, might be threatened by frequent unsupervised hybridization with Sahelian breed.

In this study, four morphotypes were found and are probably the consequence of this introgression. Furthermore, the means values of eight variables (seven linear body measurements and body weight) of the discriminant analysis revealed morphological differences among the two agroecological zones. Ranking from the tallest to the smallest, we identified Sahelian × Mossi crossed, Mossi from the Sudano-Sahelian zone, Sahelian × Djallonké crossed, and Djallonké from the Sudanian zone. The variation from large to small body size observed from the Sudano-Sahelian to Sudanian zone confirms the findings of previous studies supporting that goats body size tends to decrease following the geographical gradient north–south [8,26,31,34]. The difference in size between the native breeds of the two agroecological zones, i.e., Mossi in the Sudano-Sahelian and Djallonké in the Sudanian zones, probably comes from natural selection. Indeed, according to [31], in dry and hot areas characterized by shortages of forages during periods of drought (as is the case of goats in the Sudano-Sahelian part of Burkina Faso), animals tend to accumulate less fat because of long walks in search of forage and water and tend to be slender and thin to be able to support heat stress. On the contrary, in the humid areas where resources are relatively abundant (as is the case of goats in the Sudanian zone of Burkina Faso), animals have small body size because they have less distance to travel in search of feed and water and therefore easily take a stocky appearance. Nevertheless, the two intermediate subgroups, i.e., Sahelian × Mossi and Mossi × Djallonké, are most likely to be products of farmers’ crossbreeding practices leading to the expansion of West African long-legged goat southwards [30]. The average values obtained here for the body measurements are comparable to those found in the literature [35]. Furthermore, relatively high variation of body weight has been observed within populations in the two areas and this could provide scope for genetic improvement of goats in the two areas through selection [13].

This study showed a weak differentiation between the different genotypes with two intermediate subgroups appearing between the two well-identified breeds (Djallonké and Mossi). Animals from Mossi type and crossbreeds formed various overlapping clusters indicating low genetic differentiation. The Djallonké tend to form their group with, however, a few Mossi individuals. Genetic group assignment by hierarchical clustering based on weight and linear body measurements coincided very well with subjective breed assignment of a livestock expert, except for crossbred Sahelian × Mossi. This is consistent with the findings of previous similar studies [7,9], claiming that subjective prediction of crossbred types is harder than that of pure types.

Genes flowing from Sahelian goats into other populations of the country could explain this configuration and confirms the continuous erosion of genetic inheritance of Mossi and Djallonké breeds. Sustainable genetic improvement programs including reasonable crossbreeding need to be promoted to help farmers to meet their needs while preserving the genetic integrity of these indigenous goat breeds. Thus, different breeding programs could be suggested, including pure breeding and crossbreeding programs. To be sustainable, such breeding programs should be implemented with full participation of farmers, taking into account their preferences in animal traits and their breeding practices. Community-based breeding programs that encourage farmer participation is most likely to be fit within such a context [36,37].

## 5. Conclusions

The objective of this study was to assess phenotypic variability of local goat populations in two agroecological zones of Burkina Faso. The results showed that globally the Sudano-Sahelian population have larger body measurements than the Sudanian goat. Out of 19 morphometric traits, 8 linear body measurements and live body weight allowed the clustering of the two populations in four morphotypes, including the native Mossi breed in the Sudano-Sahelian area and the Djallonké breed in the Sudanian zone, while two hybrid groups emerged and were assigned to Djallonké × Mossi and Sahelian × Mossi crossbreds, respectively. Variation in body size was observed among these four subpopulations going from large body size in the Sudano-Sahelian morphotypes to small body size in the Sudanian ones. Several factors could explain this configuration and the variation of body size observed in these goat populations in the study areas, including adaptation and breeding practices (crossbreeding). To prevent the potential genetic erosion caused by this practice, managed breeding programs should be promoted, including supervised crossbreeding systems. Participatory breeding approach like community-based breeding programs can be suggested. In addition, the use of morphological measurements is a basis of genetic characterization of livestock populations, but this approach has some limitations because it does not provide enough information on genetic diversity or on the performance of animals in specific traits. Regarding this, our study should be completed by genomic studies using molecular markers to provide a deep understanding of the genetic makeup of goats populations in the areas.

## Figures and Tables

**Figure 1 animals-13-01931-f001:**
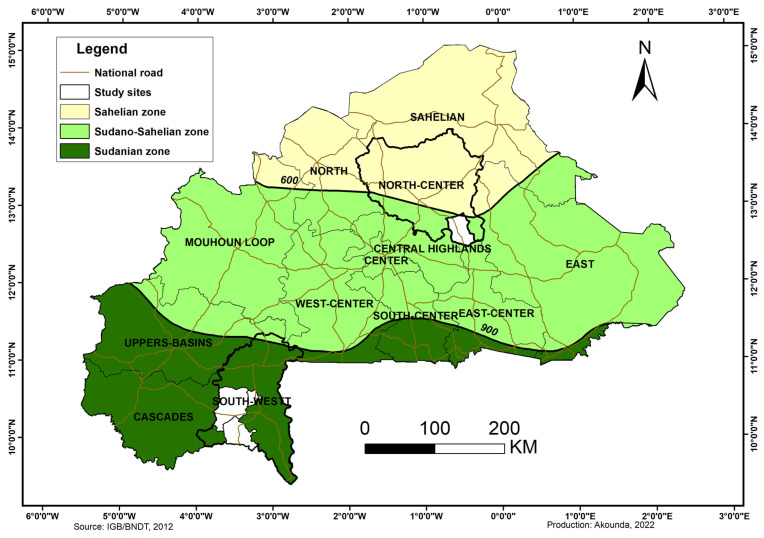
Map of Burkina Faso showing the different agroecological zones and the goats sampling areas.

**Figure 2 animals-13-01931-f002:**
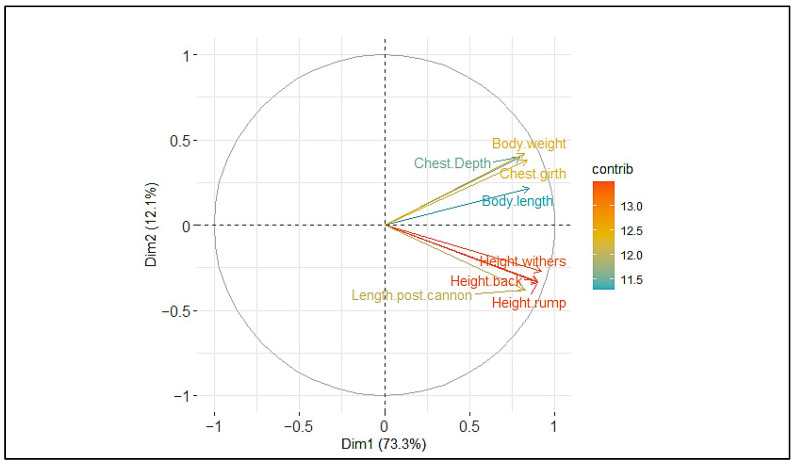
PCA of the nine body measurements of goats and their correlations with the first two axes.

**Figure 3 animals-13-01931-f003:**
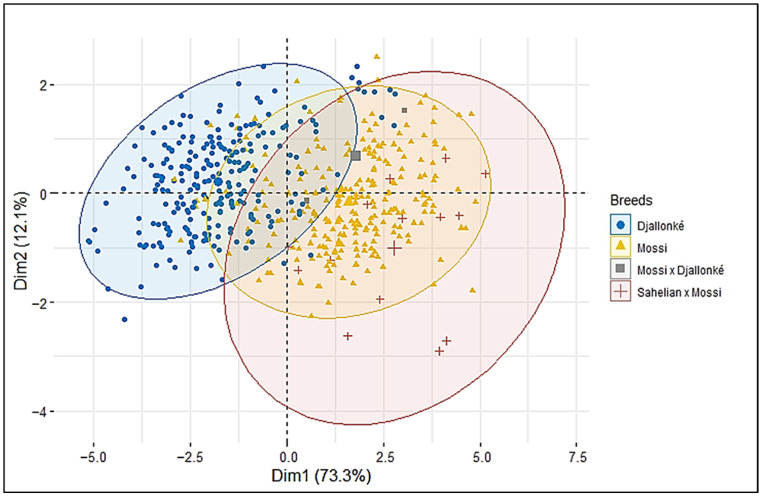
Bidimensional plot of the Principal Component Analysis (PCA) showing dispersion of the 511 goats from the Sudano-Sahelian and Sudanian zones of Burkina Faso based on nine body measurements.

**Figure 4 animals-13-01931-f004:**
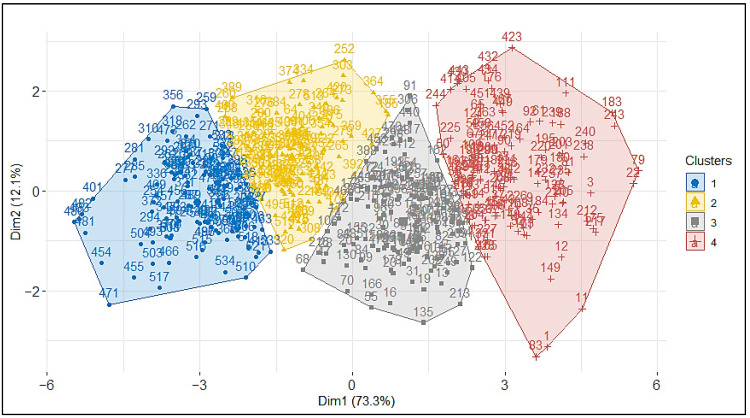
Graphic representation factor map grouping the 511 goats Sudano-Sahelian and Sudanian zones of Burkina Faso into four clusters corresponding to four morphotypes based on nine body measurements.

**Table 1 animals-13-01931-t001:** Age-group of goats sampled in two agroecological zones of Burkina Faso.

Age	Sudano-Sahel	Sudanian	Overall
3PPI	36	86	122
4PPI	150	172	322
Broken teeth	49	18	67
Total	235	276	511

**Table 2 animals-13-01931-t002:** Least squares means (LSmeans), standard errors (SE), and coefficients of variation (CV) of body weight and 19 linear body measurements of goats in Sudano-Sahelian and Sudanian agroecological zones of Burkina Faso.

Variables	Overall (*n* = 511)		SSZ (*n* = 235)		SZ (*n* = 276)		*p*-Value
	LSmean ± SE	CV	LSmean ± SE	CV	LSmean ± SE	CV	
HW	51.13 ± 0.28	6.54	55.60 ± 0.21 ^a^	5.87	47.12 ± 0.19 ^b^	7.24	<0.001
HR	51.42 ± 0.28	6.31	55.6 ± 0.20 ^a^	5.85	47.7 ± 0.20 ^b^	6.81	<0.001
HB	49.68 ± 0.27	6.27	53.87 ± 0.19 ^a^	5.90	45.92 ± 0.19 ^b^	6.66	<0.001
CD	22.89 ± 0.14	7.39	23.90 ± 0.11 ^a^	5.11	22.00 ± 0.10 ^b^	6.92	<0.001
CG	63.35 ± 0.33	6.06	65.87 ± 0.24 ^a^	6.67	61.08 ± 0.23 ^b^	8.07	<0.001
BW	21.9 ± 0.56	18.63	24.10 ± 0.26 ^a^	16.37	19.9 ± 0.243 ^b^	21.09	<0.001
PCL	15.36 ± 0.09	6.57	16.76 ± 0.06 ^a^	6.09	14.11 ± 0.06 ^b^	7.09	<0.001
ACL	10.83 ± 0.07	8.02	11.67 ± 0.05 ^a^	7.18	10.08 ± 0.05 ^b^	8.88	<0.001
BL	58.17 ± 0.31	6.33	60.60 ± 0.23 ^a^	5.51	55.99 ± 0.20 ^b^	7.09	<0.001
IW	3.81 ± 0.05	14.87	4.03 ± 0.03 ^a^	15.34	3.60 ± 0.03 ^b^	14.26	<0.001
SW	11.94 ± 0.12	11.33	12.10 ± 0.08 ^a^	12.58	11.79 ± 0.08 ^b^	10.01	0.001
TL	10.99 ± 0.11	12.02	11.56 ± 0.10 ^a^	12.11	10.47 ± 0.07 ^b^	11.89	<0.001
NL	20.37 ± 0.19	10.57	21.74 ± 0.13 ^a^	10.14	19.13 ± 0.12 ^b^	11.00	<0.001
RL	16.68 ± 0.11	7.33	17.12 ± 0.08 ^a^	6.34	16.28 ± 0.07 ^b^	8.20	<0.001
EW	9.72 ± 0.05	6.42	10.02 ± 0.04 ^a^	6.96	9.45 ± 0.03 ^b^	5.81	<0.001
EL	12.18 ± 0.09	8.93	13.55 ± 0.07 ^a^	8.78	10.96 ± 0.06 ^b^	9.01	<0.001
MC	18.20 ± 0.13	8.21	18.82 ± 0.09 ^a^	8.16	17.66 ± 0.08 ^b^	8.27	<0.001
HorL	9.06 ± 0.22	27.48	9.67 ± 0.16 ^a^	27.23	8.51 ± 0.15 ^b^	27.65	<0.001
HL	16.64 ± 0.10	6.90	17.10 ± 0.07 ^a^	6.65	16.23 ± 0.06 ^b^	7.15	<0.001

^a,b^ LSmeans with different letters in rows are significantly different at *p* ≤ 0.001, SSZ: Sudano-Sahelian zone; SZ: Sudanian zone; HW: Height at withers; HR: Height at rump; HB: Height at back; CD: Chest depth; CG: Chest girth; BW: Body weight; PCL: Posterior cannon length; ACL: Anterior cannon length; BL: Body length; IW: Ischium width; SW: Shoulder Width; TL: Tail length; NL: Neck length; RL: Rump length; EW: Ear width; EL: Ear length; HorL: Horn length; MC: Muzzle circumference; HL: Head length.

**Table 3 animals-13-01931-t003:** Average silhouette widths of the different clusters.

Cluster	Size/Number of Animals	Average Silhouette Width
1	151	0.27
2	156	0.25
3	92	0.28
4	112	0.22

**Table 4 animals-13-01931-t004:** Least squares means (LSmeans) and standard errors of body weight and seven body measurements of the four groups of goats in the Sudano-Sahelian and Sudanian agroecological zones of Burkina Faso.

Variables	Overall	Group 1	Group 2	Group 3	Group 4
BL	58.14 ± 4.36	53.34 ± 3.02 ^d^	57.18 ± 2.26 ^c^	58.81 ± 2.70 ^b^	63.16 ± 2.38 ^a^
HW	51.00 ± 5.43	45.12 ± 2.52 ^d^	47.47 ± 2.19 ^c^	53.88 ± 2.41 ^b^	57.09 ± 3.07 ^a^
HB	49.52 ± 5.03	44.09 ± 2.31 ^d^	46.23 ± 1.78 ^c^	52.58 ± 2.47 ^b^	54.75 ± 3.19 ^a^
HR	51.28 ± 5.10	45.81 ± 2.33 ^d^	48.10 ± 2.14 ^c^	54.22 ± 2.74 ^b^	56.59 ± 3.18 ^a^
CD	22.89 ± 1.93	20.77 ± 1.04 ^d^	22.62 ± 1.32 ^c^	23.13 ± 1.24 ^b^	25.03 ± 1.20 ^a^
CG	63.28 ± 4.49	57.99 ± 2.68 ^d^	62.58 ± 2.46 ^c^	64.08 ± 2.27 ^b^	68.43 ± 2.75 ^a^
PCL	15.31 ± 1.65	13.77 ± 0.89 ^d^	14.12 ± 0.97 ^c^	16.29 ± 0.93 ^b^	16.91 ± 1.08 ^a^
BW	21.89 ± 4.59	16.87 ± 1.96 ^d^	21.25 ± 2.22 ^c^	22.00 ± 2.51 ^b^	27.51 ± 3.60 ^a^

Means within the same row having different lowercase letters differ significantly (*p* < 0.05). BL: Body length, HW: Height at withers, HB: Height at back, HR: Height at rump, CD: Chest depth, CG: Chest girth, PCL: Posterior cannon length, BW: Body weight.

**Table 5 animals-13-01931-t005:** Linear discriminant analysis predictions based on body weight and linear body measurements versus subjective assignment of goat genotype.

Predict Class	Djallonké (%)	Mossi (%)	Djallonké × Mossi (%)	Sahelian × Mossi (%)	Overall (%)
Djallonké	88.93	11.07	-	-	100
Mossi	0.41	96.27	0.41	2.90	100
Djallonké × Mossi	-	-	100	-	100
Sahelian × Mossi	-	36.36	-	63.64	100

## Data Availability

Data can be made available by sending a request email to corresponding authors.

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
