# Peer review of "Morphometric Characterization of Local Goat Breeds in Two Agroecological Zones of Burkina Faso, West Africa"

_animals, 2023, doi:10.3390/ani13121931_

Round 1

Reviewer 1 Report

In my consideration, this paper is very well perforned, statistical tests are oportune, and the complete design is correct. Anyway, the general interest of this paper is limited, because only a few populations are tested inside a concrete region. 

Another subject to stand up is the quality of the redaction, it is very easy to follow the text. Quality of English looks so fine.

I think this manuscript could be aceptable after some corrections. If possible, the show of some representative photos of the implied populations could be wellcomed.

Also, I may recommend some statistical considerations. In the previous studies on goats morphological characterization it is common the use of the cannonical discriminant annalysis and the tridimensional representation of the results, it use to result very informative, so I think it could be the test of election in this research. Also  the Mahalanobis distances use to be employed to define the relations among populations, I recommend to do it in this paper.

Author Response

Response to reviewers

Dear Editor, thanks for considering our manuscript in reviewing process of your journal. We would like to thanks the anonymous reviewers for taking their time to review our manuscript. We are grateful for their comments and suggestions to improve the quality of the manuscript. We also take a note of their questions and we try to address them. New changes are tracked in the manuscript and in this response letter we also provide the new lines numbers of these changes.

Reviewer 1

Comments and suggestions to authors

In my consideration, this paper is very well perforned, statistical tests are oportune, and the complete design is correct. Anyway, the general interest of this paper is limited, because only a few populations are tested inside a concrete region.

Another subject to stand up is the quality of the redaction, it is very easy to follow the text. Quality of English looks so fine.

I think this manuscript could be aceptable after some corrections. If possible, the show of some representative photos of the implied populations could be wellcomed.

Thanks. We highly  appreciated this positive comment on our manuscript. Of course we agree with the limitation of the general interest of our paper. However we think these findings contribute to better understanding of genetic characteristics of Burkina Faso goats and more globally

Also, I may recommend some statistical considerations. In the previous studies on goats morphological characterization it is common the use of the cannonical discriminant annalysis and the tridimensional representation of the results, it use to result very informative, so I think it could be the test of election in this research. Also the Mahalanobis distances use to be employed to define the relations among populations, I recommend to do it in this paper.

Thanks for this suggestion. Indeed we tried this canonical discriminant analysis but 1 canon was obtained and it was not possible to have tridimensional representation. We assumed that this may be due that our populations are very close. And the Mahalanobis distance estimated was small and significant distance was not found.

Reviewer 2 Report

I am reviewing the manuscript entitled "Morphometric characterization of local goat breeds in two agroecological zones of Burkina Faso, West Africa," for consideration for publication in MDPI Animals.

General comments

This manuscript evaluates phenotypic differences between different goat breeds in Burkina-Faso, evaluating several morphometric traits. Some sentences are unclear, sometimes for being long, and sometimes for using conjunctions that make them hard to read. 

The method section lacks details on the statistical analysis. Moreover, it is unclear why not all traits were evaluated in the statistical analysis. 

The discussion is extremely short and incomplete. The paper has no conclusion. I suggest finishing the paper with a paragraph that wraps things up based on your aims/objective paragraph.

specific comments

Abstract: Please include all the breeds at the beginning of the abstract. It is hard to evaluate the results based on the breeds stated only with their results.

Introduction

L 36: what is number 1?

L 38: Is the 39.33% representing number of animals? Only ruminants? Please, explain

L 45: please, consider rewriting this sentence by changing how the first comma is included

L 52: This sentence is very long. Please, consider breaking it into shorter sentences. Also, I am unsure if one should consider mutation and drift evolutionary forces, as they are random by definition.

L 71: I believe the objectives/aim paragraph undersells your study. I believe the "contribute to a better understanding" term is inappropriate for this paragraph.

Materials and Methods

L 75: Is there any need to state animal care and use approval for this study?

L 84: please describe the deep changes observed objectively 

L 86: The Fulani people and the herd type relationship is unclear. Please, add more details 

L 121: how many evaluators were making those decisions? How was consistency assessed?

L 121: please rewrite the second sentence more scientifically and soundly. What is considered "good knowledge"? 

L 125 to 137: This paragraph should include many more details about your analysis. Please, expand it and explain, for instance, why only 09 body measurements were used for the HCPC. Your method section must make your research reproducible, so more details are necessary here.

Results:

L 142: define CV

L 162: How were those clusters evaluated? Having some statistical work done on that would be important, such as a kmeans or a permutation analysis. 

L 180: It is unclear if the four groups described here are related or not to the abovementioned groups.

L 185: change indeed to something else. Not clear what is the relationship between the numbers and the four group names

L 192: extremely high concordance for two groups only. I would say that 88.9 or 63 is an 'extremely high' concordance. 

Discussion

L 220: Gene flow is a strong assumption when the main similarity is due to size/weight other than shape. Please, expand that discussion.

L 224: Please, add context to the Fulani people's case, as most readers would not be familiarized with them and their history

L 238: not clear what the latter meant to refer 

L 241: please, consider rewriting this sentence for clarity 

The discussion ends prematurely, and there is no conclusion. 

Author Response

Response to reviewers

Dear Editor, thanks for considering our manuscript in reviewing process of your journal. We would like to thanks the anonymous reviewers for taking their time to review our manuscript. We are grateful for their comments and suggestions to improve the quality of the manuscript. We also take a note of their questions and we try to address them. New changes are tracked in the manuscript and in this response letter we also provide the new lines numbers of these changes.

Reviewer 2

  • General comment

This manuscript evaluates phenotypic differences between different goat breeds in Burkina-Faso, evaluating several morphometric traits. Some sentences are unclear, sometimes for being long, and sometimes for using conjunctions that make them hard to read.

Thanks. We worked to make the text more fluent

The method section lacks details on the statistical analysis.

Sorry for failing to detail the statistical analysis. We indeed use different statistical tests including ANOVA and Student-Newman-Keuls means test with the level of significance of 5% for comparison. A sentence in now added to the text about that.

Moreover, it is unclear why not all traits were evaluated in the statistical analysis.

Our objective was to see any structure between our two populations. Then we played with all traits to see their contribution to the discrimination of the subgroups. And then we choose to skip traits that not contribute to differentiate populations and keep 9 regarding their contribution to the formation of PCA axis and consequently to the separation of individuals in groups.

The discussion is extremely short and incomplete. The paper has no conclusion. I suggest finishing the paper with a paragraph that wraps things up based on your aims/objective paragraph.

Thanks for this remark. We take it into account by reinforcing our discussion and also adding conclusion.

Specific comments

Abstract: Please include all the breeds at the beginning of the abstract. It is hard to evaluate the results based on the breeds stated only with their results

Thanks. Added: Mossi breed in Sudano-Sahelian zone and Djallonké breed in Sudanian zone.

Introduction

L 36: what is number 1?

Number 1 is the first reference we cited. We mode it to the end of the sentence. Thanks

L 38: Is the 39.33% representing number of animals? Only ruminants? Please, explain

We changed it and reported it to ruminants populations (cattle, goat, sheep and camels) and this is not 43%. (Line 55-56)

L 45: please, consider rewriting this sentence by changing how the first comma is included.

This sentence has been rewritten. Thanks (Line 61 - 64)

L 52: This sentence is very long. Please, consider breaking it into shorter sentences. Also, I am unsure if one should consider mutation and drift evolutionary forces, as they are random by definition.

Sentence has been broken in 2. We hope it improve the understanding of the idea. (Line 69 -74)

L 71: I believe the objectives/aim paragraph undersells your study. I believe the "contribute to a better understanding" term is inappropriate for this paragraph.

We rephrased the sentence. Thanks. (Line 88)

Materials and Methods

L 75: Is there any need to state animal care and use approval for this study?

We thanks the reviewer for this concern. Our study involved animals, but has been done on animals in their own environment with any act that can cause suffer or pain. This has been done with the consent of farmers who participate to restrain animals.

L 84: please describe the deep changes observed objectively

Thanks for this suggestion. More details were added (Line 110 – 113)

L 86: The Fulani people and the herd type relationship is unclear. Please, add more details

The sentence added up gives more precision about the Fulani who are native of Sahelian and migrating to Sudanian zones searching for grazing areas. (Line 111 – 113)

L 121: how many evaluators were making those decisions? How was consistency assessed?

Changed: This has been made by the research team based on their experience with phenotypic appearance of local breeds. (Line 160-161)

L 121: how many evaluators were making those decisions? How was consistency assessed?

Changed: This has been made by the research team based on their experience with phenotypic appearance of local breeds. (Line 160-161)

L 125 to 137: This paragraph should include many more details about your analysis. Please, expand it and explain, for instance, why only 09 body measurements were used for the HCPC. Your method section must make your research reproducible, so more details are necessary here.

Thanks to the reviewer for suggestion. Indeed we agree that this was not very clear. In fact we plot the final PCA with 9 variables (8 body measurement + agroecological zones) for the following reasons:  We run successively PCA with all variable and we looked the quality of the PCA based on the contribution of each variables to the inertia of the two axis of the PCA. And to improve the total inertia of the PCA. And among the 19 variable, 8 were strongly correlated to the two axis in addition to the agroecological zones. To have a clear representation of our populations we decided to plot the final PCA based on the 8 body measurements and the agroecological zones. More details were provided (Line 169– 179).

Results

L 142: define CV

Defined: CV = Cœfficents of variation (Line 238)

L 162: How were those clusters evaluated? Having some statistical work done on that would be important, such as a kmeans or a permutation analysis.

Thanks. We did permutation analysis and the results are added. Table 3

L 180: It is unclear if the four groups described here are related or not to the abovementioned groups.

Yes these 4 groups described are the confirmation of the abovementioned. Hierarchical classification confirmed possible 4 subgroups.

L 185: change indeed to something else. Not clear what is the relationship between the numbers and the four group names.

Changed. The sentence was split and rephrased. (Line 299– 302)

L 192: extremely high concordance for two groups only. I would say that 88.9 or 63 is an 'extremely high' concordance.

Sorry we did not get the point

Discussion

L 220: Gene flow is a strong assumption when the main similarity is due to size/ weight other than shape. Please, expand that discussion.

Thanks. We tried to highlight this part of the discussion by adding some precisions. (Line 403 – 408)

L 224: Please, add context to the Fulani people's case, as most readers would not be familiarized with them and their history

We added more details about Fulani who migrate with their Sahelian breeds into the South leading to frequent crossbreeding between Sahelian breeds and Sudanian ones. (Line 439 - 441)

L 238: not clear what the latter meant to refer

Changed

L 241: please, consider rewriting this sentence for clarity

Thanks. We did it (475– 478).

The discussion ends prematurely, and there is no conclusion.

Thanks to the reviewer for suggesting conclusion. We now added a conclusion. (Line 548 – 567)
